# Where do Models go Wrong? Parameter-Space Saliency Maps for Explainability

**Roman Levin**[*†]
Department of Applied Mathematics
University of Washington
rilevin@uw.edu

**Manli Shu**[*]
Department of Computer Science
University of Maryland
manlis@cs.umd.edu

**Eitan Borgnia**[*]
Department of Computer Science
University of Maryland
eborgnia2@gmail.com

**Furong Huang**
Department of Computer Science
University of Maryland
furongh@cs.umd.edu

**Micah Goldblum**
Center for Data Science
New York University
goldblum@nyu.edu

**Tom Goldstein**
Department of Computer Science
University of Maryland
tomg@cs.umd.edu

## Abstract

Conventional saliency maps highlight input features to which neural network predictions are highly sensitive. We take a different approach to saliency, in which we identify and analyze the network parameters, rather than inputs, which are responsible for erroneous decisions. We first verify that identified salient parameters are indeed responsible for misclassification by showing that turning these parameters off improves predictions on the associated samples, more than turning off the same number of random or least salient parameters. We further validate the link between salient parameters and network misclassification errors by observing that fine-tuning a small number of the most salient parameters on a single sample results in error correction on other samples which were misclassified for similar reasons – nearest neighbors in the saliency space. After validating our parameter-space saliency maps, we demonstrate that samples which cause similar parameters to malfunction are semantically similar. Further, we introduce an input-space saliency counterpart which reveals how image features cause specific network components to malfunction.

## 1 Introduction

With the widespread deployment of deep neural networks in high-stakes applications such as medical imaging [20], credit score assessment [49], and facial recognition [10], practitioners need to understand why their models make the decisions they do. In fact, "right to explanation" legislation in the European Union and the United States dictates that relevant public and private organizations must be able to justify the decisions their algorithms make [46, 14]. Diagnosing the causes of system failures is particularly crucial for understanding the flaws and limitations of models we intend to employ.

---

[*]Equal contribution
[†]This work was completed prior to the author joining Amazon

36th Conference on Neural Information Processing Systems (NeurIPS 2022).

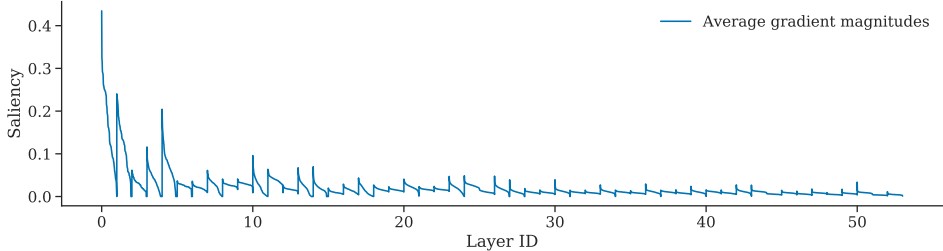

Figure 1: **Filter-wise parameter saliency profile.** ResNet-50 filter-wise saliency profile (without standardization) averaged over samples in ImageNet validation set. The filter saliency values in each layer are sorted in descending order, and each layer's saliency values are concatenated. The layers are displayed left-to-right from shallow to deep and have equal width on x-axis.

Conventional saliency methods focus on highlighting sensitive pixels [37] or image regions that maximize specific activations [13]. However, such maps may not be useful in diagnosing undesirable model behaviors as they do not necessarily identify areas that specifically cause bad performance since the most sensitive pixels may not be the ones responsible for triggering misclassification.

We develop an alternative approach to saliency which highlights network parameters that influence decisions rather than input features. These parameter saliency maps yield a number of useful analyses:

- We verify that identified salient parameters are indeed responsible for misclassification by showing that turning these parameters off improves predictions on the associated samples more than turning off the same number of random or least salient parameters.

- Nearest neighbors in parameter saliency space share common semantic information. That is, samples which are misclassified for similar reasons and cause similar parameters to malfunction are semantically similar.

- We further validate the link between salient parameters and network misclassification errors by observing that correcting a small number of the most salient parameters by fine-tuning them on *a single* sample results in error correction on *other* samples which were misclassified for similar reasons.

- By first identifying the network parameters responsible for an erroneous classification, we can then visualize the image regions that interact with those parameters and trigger the identified misbehavior obtaining interpretable insights into model mistakes and neural network's reliance on spurious correlations.

Our code for computing parameter-saliency maps is available at `https://github.com/LevinRoman/parameter-space-saliency`.

## 1.1   Related Work

**Neural network interpretability and parameter importance.** A major line of work in neural network interpretability focuses on convolutional neural networks. Works visualizing, interpreting, and analysing feature maps [52, 51, 27, 25] provide insight into the role of individual convolutional filters. These methods, together with other approaches for filter explainability [7, 56, 57] find that individual convolutional filters often are responsible for specific tasks such as edge, shape, and texture detection. Other ideas in neural network interpretability include identifying recurring patterns in model behavior [8].

The ideas of measuring neural network parameter importance has been studied in multiple contexts. Notions of neuron [34], feature [26, 35, 41, 30] and parameter [40, 47] importance have been used for AI explainability, manipulating model behavior [6], model debugging [24, 53] and pruning [1, 23, 42, 32, 50] for improving model performance.

**Input space saliency maps.** A considerable amount of literature focuses on identifying input features that are important for neural network decisions. These methods include using deconvolution approaches [52] and data gradient information [37]. Several works build on these ideas and propose

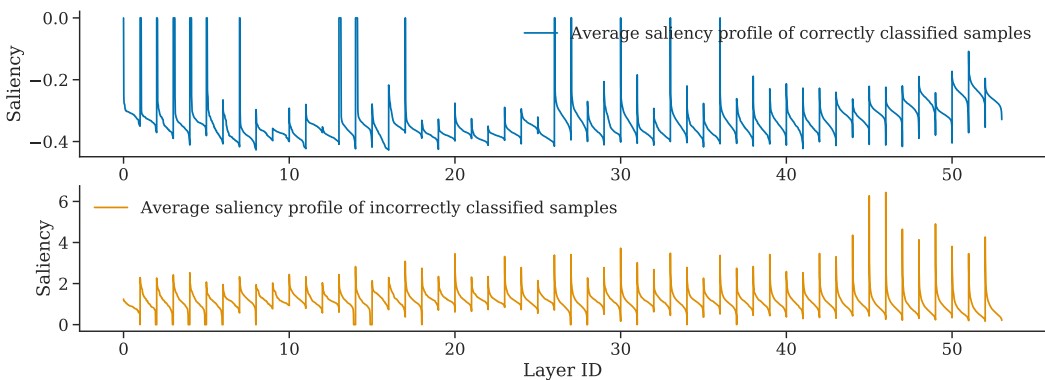

Figure 2: **Standardized filter-wise saliency profiles, correctly vs incorrectly classified samples.** Top: Standardized saliency profiles averaged over correctly classified samples in the ImageNet validation set. Bottom: Standardized saliency profiles averaged over incorrectly classified samples in the ImageNet validation set. On both panels, the filter saliency values in each layer are sorted in descending order, and each layer's saliency values are concatenated. The layers are displayed left-to-right from shallow to deep and have equal width on x-axis. Both profiles are generated on ResNet-50.

improvements such as Integrated Gradients [44], SmoothGrad [38], and Guided Backpropagation [39] which result in sharper and more localized saliency maps. Other approaches focus on the use of class activation maps [55] with improvements incorporating gradient information [34] and more novel approaches to weighting the activation maps [48]. In addition, various saliency methods are based on manipulating the input image [15, 52, 30, 3].Another line of work is aimed at evaluating the effectiveness of saliency maps and developing validation techniques [2, 4, 5, 54, 33].

Our work combines the ideas of saliency maps and parameter importance and evaluates saliency directly on model parameters by aggregating their absolute gradients on a filter level. We leverage the resulting parameter saliency profiles as an explainability tool and develop an input-space saliency counterpart which highlights image features that cause specific filters to malfunction to study the interaction between the image features and the erroneous filters.

## 2 Method

It is known that different network filters are responsible for identifying different image properties and objects [52, 51, 27, 25]. This motivates the idea that mistakes made on wrongly classified images can be understood by investigating the network parameters, rather than only the pixels, that played a role in making a decision. We develop parameter-space saliency methods geared towards identifying and analyzing neural network parameters that are responsible for making erroneous decisions. Central to our method is the use of gradient information of the loss function as a measure of parameter sensitivity.

### 2.1 Parameter Saliency Profile

Let $x$ be a sample in the validation set $D$ with label $y$, and suppose a trained classifier has parameters $\theta$ that minimize a loss function $\mathcal{L}$. We define the *parameter-wise* saliency profile of $x$ as a vector $s(x, y)$ with entries $s(x, y)_i := |\nabla_{\theta_i} \mathcal{L}_\theta(x, y)|$, the magnitudes of the gradient of the loss with respect to each model parameter. Because the gradients on *training* data for a model trained to convergence are near zero, it is important to specify that $D$ be a validation, or holdout, set. Intuitively, a larger gradient norm at the point $(x, y)$ indicates a greater inefficiency in the network's classification of sample $x$, and thus each entry of $s(x, y)$ measures the suboptimality of individual parameters.

**Aggregation of parameter saliency.** Convolutional filters are known to specialize in tasks such as edge, shape, and texture detection [51, 7, 27]. We therefore choose to aggregate saliency on the filter-wise basis by averaging the gradient magnitudes of parameters corresponding to each convolutional filter. This allows us to isolate filters to which the loss is most sensitive (*i.e.* those which, when corrected, lead to the greatest reduction in loss).

Formally, for each convolutional filter $\mathcal{F}_k$ in the network, consider its respective index set $\alpha_k$, which gives the indices of parameters corresponding to the filter $\mathcal{F}_k$ (the number of parameters corresponding to $\mathcal{F}_k$ is then given by $|\alpha_k|$). The *filter-wise* saliency profile of a sample $x$ with label $y$ is defined to be a vector $\overline{s}(x, y)$ with entries

$$\overline{s}(x,y)_k := \frac{1}{|\alpha_k|} \sum_{i \in \alpha_k} s(x,y)_i, \tag{1}$$

resulting from averaging the parameter-wise saliency profile of the sample $x$ with label $y$ on the filter level.

**Standardizing parameter saliency.** Figure 1 exhibits the ResNet-50 [18] filter-wise saliency profile averaged over the ImageNet [9] validation set, where filters within each layer are sorted from highest to lowest saliency. One clear observation is the difference in the scale of gradient magnitudes – shallower filters are more salient than deeper filters. This phenomenon might occur for a number of reasons. First, early filters encode low-level features, such as edges and textures, which are active across a wide spectrum of images. Second, typical networks have fewer filters in shallow layers than in deep layers, making each individual filter more influential at shallower layers. Third, the effects of early filters cascade and accumulate as they pass through a network.

To isolate filters that uniquely cause *erroneous* behavior on particular samples, we find filters that are abnormally salient for a sample, $x$, but not for others. That is, we further standardize the saliency profile of $x$ with respect to all filter-wise saliency profiles of $D$.

Formally, let $\mu$ be the average filter-wise saliency profile across all $x \in D$, and let $\sigma$ be an equal-length vector with the corresponding standard deviation for each entry. We use these statistics to produce the standardized filter-wise saliency profile as follows:

$$\hat{s}(x,y) := \frac{\overline{s}(x,y) - \mu}{\sigma}. \tag{2}$$

The resulting tensor $\hat{s}(x, y)$ is of length equal to the number of convolutional filters in the network, and we henceforth call it the saliency profile for sample $x$. By standardizing saliency profiles, we create a saliency map that activates when the importance of a filter is unusually strong relative to other samples in the dataset. This prevents the saliency map from highlighting filters that are uniformly important for all images, and instead focuses saliency on filters that are uniquely important and serve an image-dependent role. In the rest of the paper, unless explicitly noted otherwise, we use $\hat{s}(x, y)$ and refer to it as *parameter saliency*.

**Incorrectly classified samples are more salient.** Empirically, we observe the saliency profiles of incorrectly classified samples exhibit, on average, greater values than those of correctly classified examples. This bolsters the intuition that salient filters are precisely those malfunctioning — if the classification is correct, there should be few malfunctioning filters or none at all. Moreover, we see deeper parts of the network appear to be most salient for the incorrectly classified samples while earlier layers are often the most salient for correctly classified samples. An example of these behaviors for ResNet-50 is shown in Figure 2 which presents standardized filter-wise saliency profiles averaged over the correctly and incorrectly classified examples from the ImageNet validation set. Additionally, we note the improved relative scale of the standardized saliency profile across different layers compared to the absolute gradient magnitudes in Figure 1. Saliency profiles for other architectures could be found in Appendix A. Henceforth, we will focus specifically on saliency profiles of *misclassified* samples in order to explore how neural networks make mistakes.

## 2.2 Input-Space Saliency for Visualizing How Filters Malfunction

The parameter saliency profile allows us to identify filters that are most responsible for mistakes and erroneous network behavior. In this section, we develop an input-space counterpart to our parameter saliency method to understand which features of the image affect the saliency of particular filters. Geiping et al. [17] show that the gradient information of a network is invertible, providing a link between input space and parameter saliency space. This work, along with existing input-space saliency map tools [37, 39, 38, 55, 34], inspires our method.

Given a parameter saliency profile $\hat{s} = \hat{s}(x, y)$ for an image $x$ with label $y$, our goal is to highlight the input features that drive large filter saliency values. That is, we would like to identify image

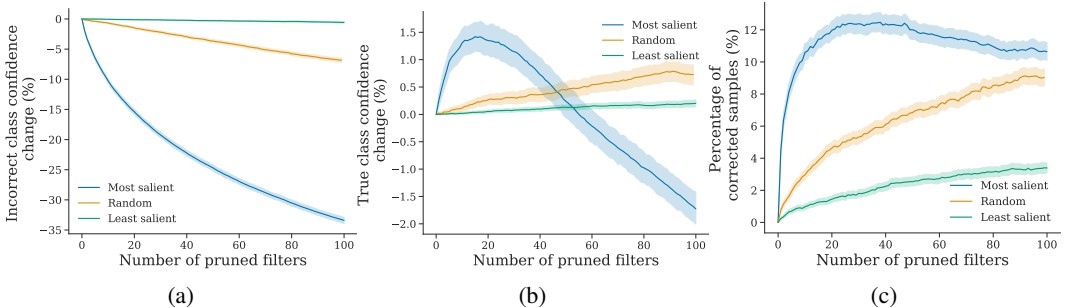

Figure 3: **Effect of turning salient filters off.** (a) Change in incorrect class confidence score. (b) Change in true class confidence score. (c) Percentage of samples that were corrected as the result of pruning filters. These trends are averaged across all images misclassified by ResNet-50 in the ImageNet validation set. The error bars represent 95% bootstrap confidence intervals.

pixels altering which can make filters more salient. To this end, we first select some set $F$ of the most salient filters that we would like to explore. Then, we create a boosted saliency profile $s'_F$ by increasing the entries of $\hat{s}$ corresponding to the chosen filters $F$:

$$(s'_F)_i = \begin{cases} \hat{s}_i, & i \notin F, \\ k\hat{s}_i, & i \in F, \end{cases} \tag{3}$$

$k > 1$ (we used $k = 100$ in our experiments). Now, we can find pixels that are important for making the chosen filters $F$ more salient and, equivalently, making the filter saliency profile $\hat{s}(x, y)$ close to the boosted saliency profile $s'_F$ by taking the following gradients:

$$M_F = |\nabla_x D_C(\hat{s}(x, y), s'_F)|, \tag{4}$$

where $D_C(\cdot, \cdot)$ is cosine distance.

The resulting input saliency map $M_F$ contains input features (pixels) that affect the saliency of the chosen filters $F$ the most.

## 3 Experiments

In this section, we aim to validate the meaningfulness of our parameter saliency method. First, we verify that salient parameters are indeed responsible for misclassification by showing on the dataset level that turning them off improves predictions on the associated samples more than turning off the same number of random or least salient parameters. This experiment is similar in spirit to removing salient features to validate input-space saliency methods [5, 33]. We then find that samples which cause similar filters to malfunction are semantically similar. We also show on the dataset level that fine-tuning a small number of the most salient parameters on a *single* sample results in error correction on *other* samples which were misclassified for similar reasons and cause similar parameters to malfunction. We then use our input-space saliency technique in conjunction with its parameter-space counterpart as an explainability tool to explore how neural networks make mistakes and how salient filters interact with visual input features.

We evaluate our saliency method in the context of image classification on CIFAR-10 [21] and ImageNet [9]. Images we use for visualization, unless otherwise specified, are sampled from ImageNet validation set. Throughout the experiments, we use a pre-trained ResNet-18 classifier [18] on CIFAR-10 and a pre-trained ResNet-50 on ImageNet. Both models are trained in a standard fashion on the corresponding dataset[34].

### 3.1 Turning Off Salient Filters

We begin validating our parameter-space saliency maps by observing the effect of turning off the salient filters. In order to remove the influence of a particular salient filter, we prune it – zero out

---

[3]https://github.com/kuangliu/pytorch-cifar (under MIT license)
[4]https://github.com/pytorch/vision (under BSD 3-Clause License)

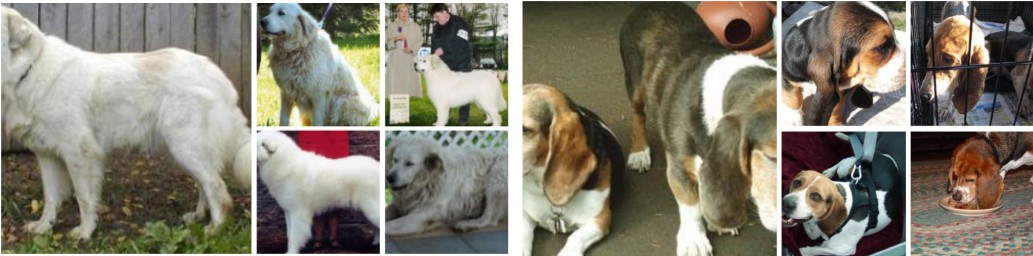

(a) Great pyrenees ↔ Kuvasz          (b) Basset hound ↔ Beagle

Figure 4: **Examples of nearest neighbors in parameter saliency space (from ImageNet)**.

the filter weights and also the biases of the associated batch normalization layers. This procedure guarantees that the corresponding input feature map to the next convolutional layer is always zero.

We gradually increase the number of pruned most salient filters and track three metrics: the change in the incorrect class confidence, the change in the true class confidence, and the percentage of the samples that flip their label to the correct class. In every case, we compare pruning the most salient filters against pruning the same number of random filters and the least salient filters. These experiments are performed on the dataset level: we average the trends across all misclassified images in the ImageNet validation set.

As shown in Figure 3, pruning the most salient filters is significantly more effective for decreasing the incorrect class confidence than random or least salient filters. Specifically, gradually pruning the top 100 salient filters achieves up to 30% drop in the incorrect class confidence score while pruning random filters yields only about 7% decrease. We also note that pruning the least salient filters does not produce any effect on the incorrect class confidence.

We repeat the same experiment with the true class confidence and observe that the highest true class confidence gain occurs when we prune around 20 most salient filters. Pruning enough salient filters eventually leads to a gradual decrease in the true class confidence. We note that this behavior is expected since we are destroying, not correcting, the inference power of all of the most sensitive filters for an image, some of which may be essential for inference. Finally, pruning random filters provides a much slower increase in the true confidence class while the least salient filters again do not produce a significant effect.

In addition, we count the number of images that were corrected as a result of pruning and find that pruning around 30 most salient filters results in the best correct classification rate of 12%. Similar to the true class confidence, the trend decreases beyond this point. Pruning random filters increases the percentage of corrected samples at a much slower rate and does not perform better than the most salient filters when pruning up to 100 filters. Notably, pruning the least salient filters manages to correct a nontrivial number of samples but still much smaller than pruning random filters.

These experiments suggest that the identified salient parameters are indeed responsible for misclassification since turning these parameters off improves predictions on the associated samples more than turning off the same number of random or least salient parameters.

### 3.2    Nearest Neighbors in Parameter Saliency space

We validate the semantic meaning of our saliency profiles by clustering images based on the cosine similarity of their profiles. In this section, we present visual depictions of a nearest neighbor search among all images in the ImageNet validation set. We also conduct this analysis on CIFAR-10 images, and this can be found in Appendix A.

We find that the nearest neighbors of misclassified images in saliency space are mostly other misclassified images from the same pair of predicted and true classes but possibly in reverse order. For example, in Figure 4, the reference image in (a) is a great Pyrenees misclassified as kuvasz, and the 4 images with the most similar profiles exhibit either the same misclassification or the reverse (i.e., kuvasz misclassified as great Pyrenees). Intuitively, the common salient parameters across these neighbors are those which are important for discriminating between the two classes in question but are not well-tuned for this purpose.

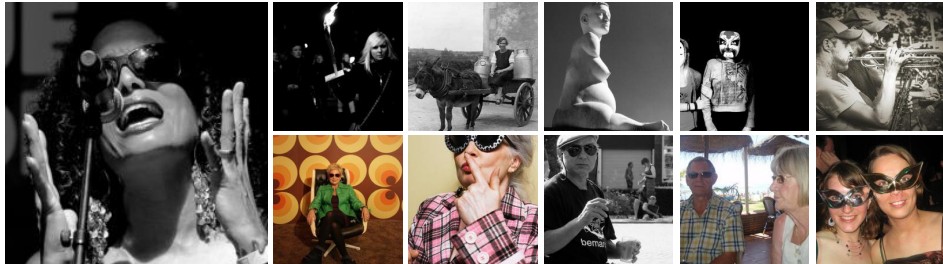

Figure 5: **Neighbors in parameter saliency space found using only early or only deep layers.** The reference image is in the first column. Images in the top row resemble the reference image in the saliency on early layers of VGG-19, and images in the bottom row are found using deeper layers.

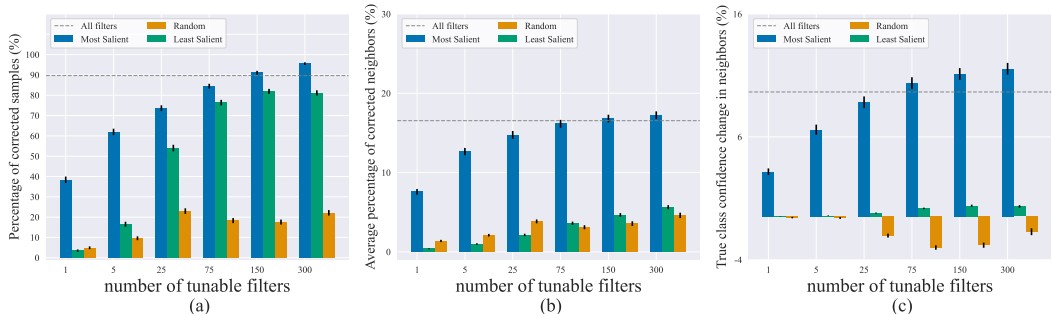

Figure 6: **Effect of updating a limited number of filters.** (a) Percentage of misclassified samples corrected after fine-tuning. (b) Average percentage of nearest neighbors that are also corrected after fine-tuning. (c) Average change in the confidence score of the true class among nearest neighbors. The horizontal dashed line in each plot is the effect of updating the entire network.

Note that we find the concept of "being similar" in parameter saliency space to be different from the one in image space. The nearest neighbors we find are often not similar in a pixel-wise sense, but rather they are similar in their reason for causing misclassification. For example, images in Figure 4 (b) are beagles mistaken by a network for basset hounds and vice versa. We find that these pictures are either taken from a high angle or do not include the dog's legs, making the leg length, a major distinction between the two breeds, indistinguishable from the picture. We include more example images along with their nearest neighbors in Appendix A.

In addition, we compute nearest neighbors when only considering filters in a specific range of layers in order to visualize the types of misbehavior triggered by network components (filters) at various network depths. We search for similar images using parameter saliency in the shallow and deep layers of a VGG-19 network [36], which we divide into the shallow and deep parts that respectively occur up to and after layer `relu4_1`. The top row of figure Figure 5 shows neighbors found using shallow parameters, which share basic image attributes such as color histogram, while images in the bottom row share more abstract similarities.

### 3.3 Correcting Mistakes by Fine-Tuning Salient Filters

To validate that salient filters are more responsible for the erroneous behavior of neural networks, we show that updating a limited number of salient filters on a single misclassified image can correct the mistakes made by a neural network on other images that were misclassified because of similarly malfunctioning filters – on nearest neighbors of that image in the parameter saliency space. In this experiment we fine-tune a pretrained ResNet-50 for one step on a single image for which the network makes a wrong prediction and track the model's performance on the image's nearest neighbors found through the process introduced in Section 3.2. While filter saliency profiles are standardized with respect to the ImageNet validation set, we find the nearest neighbors from an independent test set – ImageNet-v2, collected by Recht et al. [31] separately from the original ImageNet data. This way, the label information of the nearest neighbors is not used in fine tuning in any way.

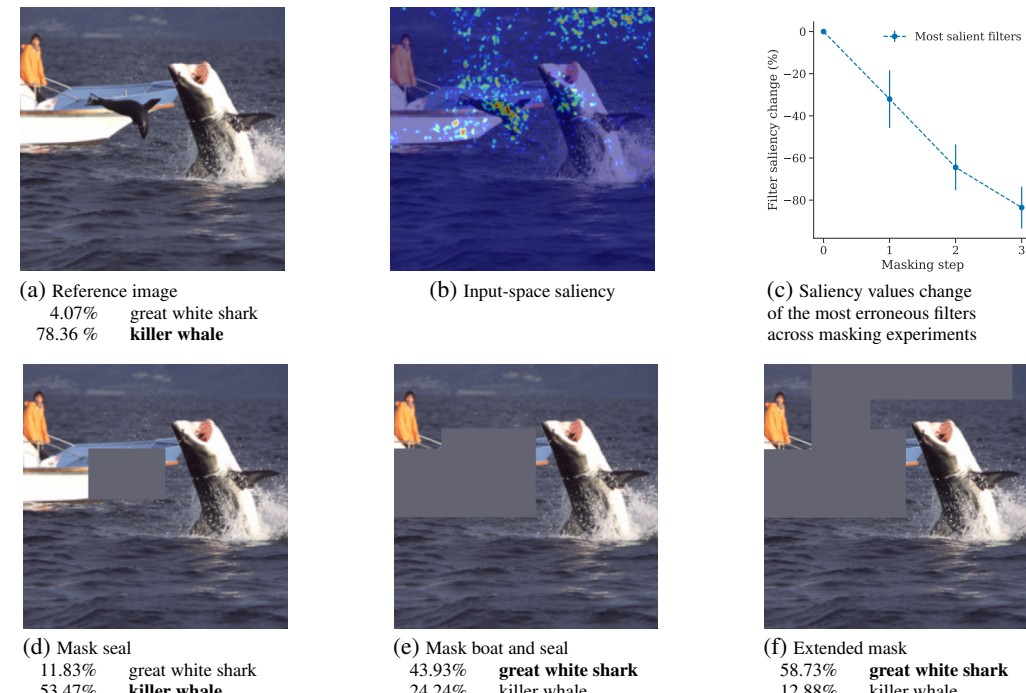

Figure 7: **Interaction between input features and salient filters.** (a) Reference image of "great white shark" misclassified by ResNet-50 as "killer whale" with confidence scores. (b) Input-space saliency visualization. Pixels that cause the top 10 salient filters to have high saliency. (c) Change in saliency values of the erroneous filters across masking experiments. The vertical bars represent the standard deviation of the change across 10 most salient filters. (d)-(f) Masking experiments.

We restrict the number of tunable filters to be no more than 1.0% of the total number of filters in the network, and we update the chosen filters by taking one step of gradient descent with a normalized step. We compare fine-tuning the most salient filters with two other choices of tunable filters: the least salient filters and random filters. We use misclassified images from the ImageNet validation set, making up to over 10,000 samples. We evaluate the effect of fine-tuning on each of these images independently and find the corresponding nearest neighbors from a holdout ImageNet-v2 test set where we limit the search scope exclusively to misclassified images too.

In Figure 6, we compare the average performance of our three choices of tunable filters under three metrics. Panel (a) presents a the percentage of samples that are corrected after fine-tuning as a sanity check and we see that updating 150 most salient filters (~0.6% of total filters) is enough to achieve the same percentage of corrections as updating the entire network.

The second and third metrics evaluate the effect on the nearest neighbors of the training sample (Figure 6 (b),(c)). Note that for a given training sample, its nearest neighbors are not involved in our one-step single sample fine-tuning process. By tracking model predictions and true class confidence scores among the 10 nearest neighbors of each sample, we find that fine-tuning salient filters is significantly more effective than other choices. Results in Figure 6, (b) and (c), also imply that the nearest neighbors found using our method are the images that are wrong for similar reasons and that they can be corrected altogether by only updating the salient filters on a single image. We note that we do not propose a new pruning or fine-tuning method. Rather, we use these experiments to verify that the salient filters are indeed responsible for misclassification.

## 3.4 Interpretable Network Failures: Visualizing Input Features Which Trigger Filter Malfunctions

**Case study.** To present an example of how our approach could be used as an explainability tool, we begin this section with a case study of an image misclassified by ResNet-50 as "killer whale" (Figure 7(a)). The correct label of the image is "great white shark". Our goal is to study the interaction between

the most salient filters and input features. We first identify filters most responsible for misclassification by computing the filter saliency profile and visualize parts of the image that drive the high saliency values for those filters using the input-space saliency counterpart (as introduced in Section 2.2).

Panel (b) of Figure 7 presents our image-space visualization for the ten most salient filters – the pixels which trigger misbehavior in these filters are highlighted. For example, we see that the seal and boat are both triggers. One natural hypothesis is that the seal looks like a killer whale to the network and is the source of the classification error. We check this by masking out the seal (Figure 7 (d)) . However, although the probability of "killer whale" goes down and the probability of the correct class increases, the network still misclassifies the image as "killer whale".

Now, if we mask out exactly the most salient areas of the image according to our visualization (see Figure 7 (b), (e)), the network manages to flip the label of the image and classify it correctly. If we extend our mask to the less pronounced, but still salient, areas of the image as in Figure 7 (f), we observe that the correct class confidence increases even more while the probability of the incorrect "killer whale" label further decreases. Additionally, we find that masking out the non-salient parts of the image results in even worse misclassification confidence than that of the original image (see Appendix A). In order to further investigate the effect of the salient region, we pasted it from this image onto other great white shark images (see Appendix A) and observed that this drives the probability of "killer whale" up for 39 out of 40 examples of great white sharks from the ImageNet validation set with an average increase of 3.75%.

Our experiments suggest that secondary objects in the image are associated with the misclassification. However, we see that the erroneous behavior of the model does not just stem from classifying a non-target object in the image. It is possible that the model correlates the combination of sea creatures (e.g. a seal) and man-made structures (e.g. a boat) with the "killer whale" label. We note that images of killer whales in ImageNet often have man-made structures which look similar to the boat (see Appendix A for examples).

Finally, at each step of our masking experiments, we recompute the saliency values of the originally chosen 10 filters (i.e. the filters that caused erroneous behavior on the reference image). From Figure 7 (c), we observe that as we mask out the input features according to our input-saliency, the saliency values of those filters decrease gradually and reach an 80% drop, confirming that highlighted regions indeed drive the high saliency of the chosen filters.

**Dataset-level masking experiment.** We performed the experiment of masking the non-foreground image areas on the dataset level and observed similar trends to the case study. Specifically, we trained a ResNet-50 on half of the ImageNet train set, and then used the incorrectly classified images from the other half of the train set with target object bounding boxes for the experiment (we used annotations from the ImageNet website[5]). We computed our input saliency for every image and masked out the most salient regions of those images while preserving the target object. We observed that as a result of masking, the softmax output corresponding to the correct class increases ($p < 0.05$) on average and decreases for the incorrect class ($p < 0.05$), while masking the same number of randomly selected pixels (similarly outside of the target object bounding box) does not produce a statistically significant change in either correct or incorrect class confidence scores ($p = 0.854$ and $p = 0.695$, respectively).

Figure 8 showcases input space visualizations with different illustrative examples of neural network mistakes. For a thorough discussion of mistake categories we find using our method, we refer to Appendix A.9.

# 4   Discussion

Numerous applications demand that practitioners be able to understand the decisions their models make, especially when those decisions are incorrect. Existing methods for explainability focus on locating the input regions to which the network's output is sensitive or on associating network components with specific roles. In contrast, we develop a framework for finding the exact filters which are responsible for faulty predictions and studying the interactions between these filters and images. This direction yields an interpretable understanding of model behaviors.

---

[5]https://image-net.org/download-bboxes.php

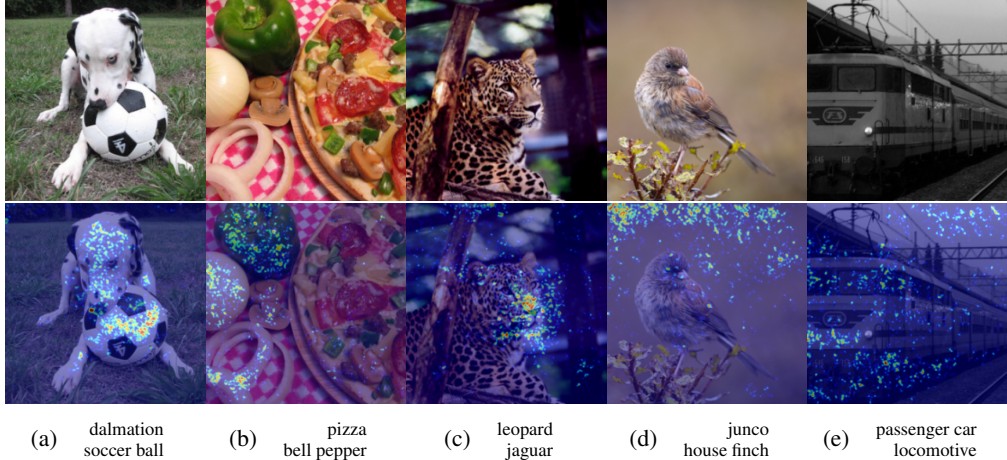

| (a) | dalmation
soccer ball | (b) | pizza
bell pepper | (c) | leopard
jaguar | (d) | junco
house finch | (e) | passenger car
locomotive |

Figure 8: **Different types of network mistakes.** All of the presented images are misclassified by ResNet-50. The correct class label is specified in the top row and the incorrect class label – in the bottom row of the subcaption on each panel. (a)-(b) The target object is confused with another object in the image. (c) A regular mistake. The salient pixels are focused on the target object features which confuse the network. (d) Background features confuse the network. (e) An example of a noisy label where the network is "more correct" than the target label. These are examples where masking top 5% of the salient pixels corrects the misclassification.

# 5   Acknowledgments

This work was supported by the DARPA YFA program, DARPA GARD, the ONR MURI program, the National Science Foundations Division of Mathematical Sciences, the AFOSR MURI program, and Capital One Bank.

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
