# OpenReview forum: "Where do Models go Wrong? Parameter-Space Saliency Maps for Explainability"
_NeurIPS.cc/2022/Conference — NeurIPS 2022 Accept_

### Official Review · Reviewer_pViJ · 2022-07-10

**Rating:** 6
**Confidence:** 4
**Soundness:** 3 good
**Presentation:** 2 fair
**Contribution:** 3 good

**Summary:**

An interesting paper to establish a link between parameter space saliency and input saliency for the purpose of diagnosis of misclassification in deep NNs. The paper demonstrates that by turning salient filters on/off the prediction of NNs could be corrected.

**Questions:**

- How does the saliency in input space correlate to the true image label?
- Could it be the findings are more relevant to datasets than models?

**Limitations:**

It would be interesting to see if similar results could be obtained with other NN architectures.

**Strengths And Weaknesses:**

The paper addresses an interesting question that is to study the relation between saliency and network predictions. The approach is thought provoking. Nonetheless, the paper is  vague about the details when it comes to saliency profile calculation and could have a better representation. The experiments are limited but convincing with respect to the conclusion. There is, however, an overlooked area, in my opinion. That is the dataset bias and label noise, when the most visually salient input is not the same as the image label. It remains unanswered what really goes wrong, is it hidden in the data or in the model or both!?

---

> ### Author Response · Authors · 2022-08-02
> **Response to Reviewer pViJ**
>
> Dear Reviewer pViJ,
>
> We thank you for your time and feedback.
>
> Regarding your questions on the interaction of data biases, label noise, model mistakes and parameter saliency, we note that our case study and dataset-level masking experiments from section 3.4 exactly address what goes wrong “when the most visually salient input is not the same as the image label”. Namely, when features in training data are spuriously correlated with a target label, the model may learn to associate the spurious feature with the correlated label, and our method allows us to highlight these spuriously correlated features. Taking our case study as an example (Figure 7), we observe that man-made structures like the boat may be correlated with the “killer whale” label or there could be spurious background features like the highlighted areas of the sky. Our method exposes these spurious correlations and masking these spurious features increases the correct class confidence and decreases the incorrect class confidence. Upon inspection of the training data for the “killer whale” label, we indeed observe that man-made structures often appear since the “killer whale” label actually corresponds to orcas that frequently appear in swimming pools (please, see section A.9 of the Appendix, Figure 20 for more details). To answer your question on what goes wrong – in the case study, the model goes wrong because it erroneously relies on spurious correlations, while this erroneous behavior is likely related to the biases in the training data.
>
> Regarding the results for other NN architectures, please see section A.3 of the Appendix for parameter saliency profiles of other architectures and section A.5 for the fine-tuning experiments with VGG-19.
>
> Additionally, we clarified the parameter saliency profile calculation and added more details, please see the changes highlighted in blue in our updated draft. We hope this addresses your concern about the presentation quality and we would be happy to work with you to clarify any parts of the paper that you may still find unclear.
>
> Thank you again for your feedback, we hope that our response addresses your questions and we would appreciate it if you would consider raising your score in light of our response. Please let us know if you have any additional questions we can address.

---

> > ### Comment · Reviewer_pViJ · 2022-08-10
> > **Thanks**
> >
> > Thank you for answering my questions and concerns. I think the paper is a good one; it could further be discussed and open some avenues in the community and raise my score.

---

> ### Author Response · Authors · 2022-08-08
> **Following Up with Reviewer pViJ**
>
> Thank you again for your thoughtful review. Does our response help address your feedback? We would appreciate the opportunity to engage further if needed.

---

### Official Review · Reviewer_PwHG · 2022-07-12

**Rating:** 7
**Confidence:** 4
**Soundness:** 4 excellent
**Presentation:** 4 excellent
**Contribution:** 3 good

**Summary:**

"Where do Models go Wrong" presents a saliency method that highlights model filters important to the output. The paper validates these parameters are salient by turning off the salient parameters on misclassified predictions and showing an increase in corrected classification. They evaluate the semantic meaning of parameter space saliency by quantifying each input by its parameter saliency and analyzing the resulting space. The paper finds neighbors often share misclassification causes instead of visual similarities, showing that parameter saliency captures patterns in model behavior. Parameter saliency can also identify filters to fine-tune as a way to correct model mistakes. Finally, the paper shows how to use parameter saliency to analyze an image in an explainability task by masking various portions of the image related to salient parameters.

**Questions:**

* The proposed parameter saliency method computes the gradient of the loss with respect to each model parameter. This seems equivalent to Vanilla Gradients applied to model parameter space. Is it possible to use variants of other input-space saliency methods to compute parameter-space saliency (e.g., SmoothGrad)? If so, have you tried any, and how did they compare?
* Could you use parameter saliency to prune models for model compression? Have you tried this or compared it to other pruning techniques?
* Can you provide a similar analysis for correctly classified inputs? It would be valuable to confirm that pruning salient filters decreases model output confidence for correctly classified images. Similarly, it would be interesting to see if the nearest neighbors of correct classifications are other correct classifications. Are neighbors from correctly classified images more likely to be in the same class, or would we find neighbors with similar confidence in the output class?

**Limitations:**

The appendix provides an interesting and thorough discussion about adversarial attacks, a comparison to GradCAM, and limitations. I would suggest moving at least some of this to the main text as it provides compelling context and considerations for the method.

**Strengths And Weaknesses:**

**Strengths**
* *Experimental baselines* --- The paper evaluates the saliency map by pruning salient filters and measuring the change to the model's output. They compare this procedure to pruning random filters and pruning the least salient filters. These comparisons show a significant difference in the model's output when salient filters are pruned.
* *Thorough evaluation* --- The paper validates their results on various models architectures trained on image and language modalities. They also test their saliency method using existing model randomization tests and find it model dependent.
* *Clarity* --- The paper is very well written and precise.

**Weaknesses**
* *Analysis of other saliency methods* --- The proposed parameter saliency method computes the gradient of the loss with respect to each model parameter and aggregates the gradients per filter. While other methods could compute parameter saliency, the paper does not include evaluations of any alternatives. It would be informative to see how consistent the parameter saliency is between methods. For instance, SmoothGrad could be used in this method by perturbing the input features but taking the gradient with respect to the model parameter.
* *Analysis of correctly classified samples* --- The method focuses on identifying filters contributing to misclassification. While this is undoubtedly important to identify and fix model errors, it is equally important to understand why the model made a correct classification. For example, it is critical to know why a model correctly predicted disease in a medical setting.

**Related Work**
* *Model error patterns* --- Figure 8 classifies different types of mistakes based on where the saliency is with respect to the main object. "Shared Interest: Measuring Human-AI Alignment to Identify Recurring Patterns in Model Behavior" by Boggust et al. has a similar metric for identifying and measuring these patterns. You might consider citing this method or using it to compute the Shared Interest of the different mistake categories you identified.
* *Saliency method evaluation* --- I appreciate that you evaluate your saliency method via the model parameterization test in the appendix. You might also consider discussing or validating your input-wise saliency on other proposed tests like the pointing game ("Top-down Neural Attention by Excitation Backprop" by Zhang et al.)
* *Removing salient parameters experiments* --- The experiment removing saliency parameters to validate parameter-space saliency is analogous to removing salient features to validate input-space saliency. These validation approaches have been proposed in "Explaining recurrent neural network predictions in sentiment analysis" by Arras et al. and "Evaluating the visualization of what a deep neural network has learned" by Samek et al.

**Minor Issues**
* Figures 3, 6, and 7 are too small and hard to read.
* Please move figures closer to their corresponding text. Having figures pages away from the reference text is challenging to read.

---

> ### Author Response · Authors · 2022-08-02
> **Response to Reviewer PwHG**
>
> Dear Reviewer PwHG,
>
> Thank you for your supportive review and thorough feedback!
>
> Regarding your first question on using SmoothGrad for parameter saliency, thank you for suggesting an interesting experiment! Prompted by your review, we have now implemented SmoothGrad for parameter saliency and added the results to Section A.1 of the updated Appendix. Please, see Figure 9 in our updated Appendix. To summarize, we observe that our original Vanilla-like parameter saliency is consistent with SmoothGrad-based parameter saliency both in terms of the parameter saliency profile and the input-space saliency. On our case study “great white shark” example, the two methods produce very similar results for small amounts of noise injected by SmoothGrad into the input image (up to 10% noise, while the network similarly misclassifies both the original and perturbed input image as “killer whale”). However, as the amount of SmoothGrad noise increases, the spurious features such as the background features in the sky get highlighted less, and the input-space saliency becomes more focused on the seal. This behavior is expected since noise corrupts the sparse spurious correlations the network relied on before while another misclassification reason – the seal – remains present. As the amount of SmoothGrad noise increases further (to 50% noise), the network misclassifies the perturbed input image with labels other than “killer whale” (e.g. “mongoose”, “plane”) and the input-space saliency stops being consistent with the Vanilla-like original approach.
>
> Regarding your second question on using parameter saliency for pruning with the objective of model compression, we have not explored this direction as we decided that such an objective was outside the scope of this paper. However, this is an interesting idea, we note that parameter saliency may indeed be leveraged for model compression by pruning the least salient model parameters.
>
> Regarding your third question, prompted by your review, we ran new experiments and we now provide the analysis for correctly classified samples, please, see section A.2 of the updated Appendix. Our new experiments confirm that pruning salient filters decreases model output confidence for correctly classified images (please see Figure 10 in Appendix). For the nearest neighbors of correct classifications, we find that on average, 89.0% of the 10 nearest neighbors of the correct inputs are also correct, whereas for incorrectly classified samples, only 10.8% of their 10 nearest neighbors are correctly classified. Please, see section A.2 of the updated Appendix for additional details.
>
> Regarding the minor points, we made Figures 3, 6, and 7 larger and moved all the figures closer to the corresponding text. We also added citations for all of the suggested related works, thank you for pointing them out! Regarding your suggestion to move some of the Appendix sections to the main body, we would be happy to do that using the additional page for the camera-ready version. We envision adding to that page a section on alternative ways to compute parameter saliency: the SmoothGrad-like approach and the discussion on parameter-space adversarial attacks.

---

> > ### Comment · Reviewer_PwHG · 2022-08-07
> > **Response to Revisions**
> >
> > I want to thank the authors for their thorough response to my review. The additional experiments and changes to the paper look good to me. I continue to think this paper should be accepted.

---

### Official Review · Reviewer_ywxe · 2022-07-13

**Rating:** 5
**Confidence:** 3
**Soundness:** 3 good
**Presentation:** 3 good
**Contribution:** 2 fair

**Summary:**

In this paper, the authors analyze salient network parameters, which are responsible for erroneous decisions. They conduct experiments to find salient parameters responsible for faulty predictions and show the relation between these parameters and regions in the input image.

**Questions:**

How is this paper different from adversarial example?

**Limitations:**

I believe there would be no potential negative societal impact.

**Strengths And Weaknesses:**

Strengths: Instead of focusing on saliency maps like other work, this paper pays attention to salient network parameters, and the findings are interesting.

Weaknesses: I am not expert in this research field. However, I have a concern about the novelty of this paper. Parameter saliency is not new, and it’s already done in adversarial example research. The authors also do not clarify differences between this paper and adversarial example.

---

> ### Author Response · Authors · 2022-08-02
> **Response to Reviewer ywxe**
>
> Dear Reviewer ywxe,
>
> Thank you for your time and feedback.
>
> Regarding your request to clarify the differences between our paper and adversarial examples, we first note that classical adversarial examples [1] are small image perturbations which exploit neural network sensitivity to input features and therefore do not offer a direct measure of or proxy for parameter saliency. Instead, input-space adversarial examples are naturally related to classical input-space saliency maps [2,3] which similarly leverage the network sensitivity to input features.
>
> However, similarly to how input-space saliency maps are naturally related to input-space adversarial attacks, parameter-space adversarial attacks (used e.g. in optimizers to find flat loss minima [4,5,6]) are indeed related to our parameter saliency method. In fact, we already discussed this relationship in Section A.7 of the Appendix which outlines the similarities and differences with our work and presents experiments comparing the two approaches. To summarize the results, while our method is more computationally efficient and hyper-parameter free, we find that saliency profiles generated using the more complex approach of parameter-space adversarial attacks instead are similar to our method (with 0.97-0.99 cosine similarity between the parameter saliency profiles generated by our method and by adversarial attacks). Importantly, parameter-space attacks have been used to find flat loss minima and for improving robustness, while we instead apply the ideas of parameter saliency for the identification of erroneous parameters and explainability.
>
> Thank you again for your feedback, we hope that our response addresses your questions and we would appreciate it if you would consider raising your score in light of our response. Please let us know if you have any additional questions we can address.
>
> [1] Goodfellow, I.J., Shlens, J. and Szegedy, C., 2014. Explaining and harnessing adversarial examples. arXiv preprint arXiv:1412.6572.
>
> [2] Simonyan, K., Vedaldi, A. and Zisserman, A., 2013. Deep inside convolutional networks: Visualising image classification models and saliency maps. arXiv preprint arXiv:1312.6034.
>
> [3] Selvaraju, R.R., Cogswell, M., Das, A., Vedantam, R., Parikh, D. and Batra, D., 2017. Grad-cam: Visual explanations from deep networks via gradient-based localization. In Proceedings of the IEEE international conference on computer vision (pp. 618-626).
>
> [4] Foret, P., Kleiner, A., Mobahi, H. and Neyshabur, B., 2020. Sharpness-aware minimization for efficiently improving generalization. arXiv preprint arXiv:2010.01412.
>
> [5] Kwon, J., Kim, J., Park, H. and Choi, I.K., 2021, July. Asam: Adaptive sharpness-aware minimization for scale-invariant learning of deep neural networks. In International Conference on Machine Learning (pp. 5905-5914). PMLR.
>
> [6] Du, J., Yan, H., Feng, J., Zhou, J.T., Zhen, L., Goh, R.S.M. and Tan, V.Y., 2021. Efficient sharpness-aware minimization for improved training of neural networks. arXiv preprint arXiv:2110.03141.

---

> ### Author Response · Authors · 2022-08-08
> **Following Up with Reviewer ywxe**
>
> Thank you again for your thoughtful review. Does our response help address your feedback? We would appreciate the opportunity to engage further if needed.

---

### Official Review · Reviewer_N8GQ · 2022-07-14

**Rating:** 9
**Confidence:** 3
**Soundness:** 4 excellent
**Presentation:** 4 excellent
**Contribution:** 4 excellent

**Summary:**

This paper approaches the topic of network explainability and visualization with a novel approach for isolating the factors contributing to a model's behaviour that focuses on network parameters rather than the previously more common approach of identifying image regions of highest impact. This is a subtle but non-trivial change, and leads to a number of useful new analyses of network behaviour, and in particular shows great promise in selectively training network parameters to correct errors.

**Questions:**

I do not have any questions beyond the minor points raised above. The paper was overall clearly written.

**Limitations:**

I do not foresee any negative societal impacts as a direct result of this work.

**Strengths And Weaknesses:**

Strengths:
This paper presents a novel idea that is intuitive (and almost obvious in retrospect, but many great ideas seem so!) and quite broad in application and effect. The experiments that are performed are easy to follow and clearly developed, offering strong evidence for the potential of this method, particularly in terms of targeted tuning and updating of network behaviour.

Weaknesses:
- Perhaps I missed it, but what is the effect on network behaviour for objects that are not neighbours to the misclassified sample? Are new errors introduced (and if so, at what rate? Is this still a strong net benefit, or are we trading one set of errors for another)?
- The visualization aspect of this method appears to essentially still end up looking similar to class activation maps in methods like Grad-CAM (i.e. it gets projected down to a pixel-wise map showing image regions that are most affecting the model behaviour on the given sample). While I am not sure that this is in and of itself a major weakness in the method (it is an intuitive visualization technique), the lack of comparison between the visualizations produced by the proposed method with prior image region based methods (e.g. Grad-CAM) makes it harder to evaluate their role and effectiveness.

---

> ### Author Response · Authors · 2022-08-02
> **Response to Reviewer N8GQ**
>
> Dear Reviewer N8GQ,
>
> Thank you for your supportive review and the kind comments about our work, we are delighted that you found the idea of parameter saliency intuitive! We also thank you for the insightful questions, and we address your feedback below.
>
> Regarding comparison to other visualization tools, in fact, we already compared our method to GradCAM and presented the results in Appendix (section A.8). Here we summarize the discussion of that section. While GradCAM is not specifically geared towards the goal of highlighting image regions which drive erroneous behavior as our method, using GradCAM with predicted label can be leveraged to explain where in a misclassified image the network sees the object corresponding to the predicted wrong label. To use the “great white shark” image misclassified as “killer whale” from Figure 7 as an example, our method answers the question of “Why is this not the great white shark?” while GradCAM answers the question of “Why is this a killer whale?”. Additionally, GradCAM was designed to be highly localized to the object corresponding to the label of interest, while our approach highlights sparse fine-grained features of images which we believe is a desirable property for our specific application as it helps expose spurious correlations that the network relies on when it makes a mistake. In the comparison, we indeed see that GradCAM is always focused on the target object while our method often highlights sparse background features (an example would be the spurious background features – the sky and the boat – highlighted in Figure 7 while GradCAM is focused on the shark). For more details of our comparison, please see Figures 22 and 23 in section A.10 of the updated Appendix.
>
> Regarding if our framework could be leveraged to improve the overall accuracy, this is a good idea, but the focus of our work is on explainability and visualization.  While we do see new errors introduced as the result of our fine-tuning experiment, so that as you said we trade errors for errors, we note that the primary goal of the experiment was to validate that salient filters are responsible for the erroneous behavior of neural networks by showing that updating a limited number of salient filters on a single misclassified image can correct the mistakes made by a neural network on other images that were misclassified because of similarly malfunctioning filters – the nearest neighbors in parameter saliency space. For targeted tuning, even though we trade one set of errors for another, this is still useful since some errors might be higher cost than others in applications where machine learning models are expected to conform to certain guardrails (e.g. for correcting socially inappropriate image classifier or language model outputs).

---

> > ### Comment · Reviewer_N8GQ · 2022-08-10
> > **Brief Response**
> >
> > Thank you for engaging with my review and I appreciate your substantive explanations to my questions. I still think this is a strong paper that should be published.
> >
> > My only additional comment is that I think that this issue of trading one set of errors for another should be mentioned in the paper (possibly under the discussion or in the context of future directions to explore with this method), because otherwise this may catch a user of the method off guard if they have not fully thought this through or naively attempt to apply it to tune their model.

---

### Meta-Review · Area_Chair_ueb3 · 2022-08-25

**Recommendation:** Accept
**Confidence:** Certain

**Metareview:**

This work was already appreciated in its original version by all reviewers: it presents a simple novel idea, intuitive, and with high impact for the community and application-wise. The paper was considered well written and clear, methodological and experimental/evaluation parts are well developed and further improved after rebuttal, also providing additional experimental evidences. Some issues and requests of clarifications were of course highlighted by the reviewers, and the authors feedback was more than adequate, fully satisfying the raised concerns.
In the end, after rebuttal, all reviewers confirm the validity of the work and its acceptance to NeurIPS 2022.




**Award:**

Yes

---

### Decision · Program_Chairs · 2022-09-14

Accept